# Isolation of Methane Enriched Bacterial Communities and Application as Wheat Biofertilizer under Drought Conditions: An Environmental Contribution

**DOI:** 10.3390/plants12132487

**Published:** 2023-06-29

**Authors:** Adoración Barros-Rodríguez, Carlos García-Gálvez, Pamela Pacheco, Marina G. Kalyuzhnaya, Maximino Manzanera

**Affiliations:** 1Institute for Water Research and Department of Microbiology, University of Granada, 18071 Granada, Spainpamelapachecod@gmail.com (P.P.); 2VitaNtech Biotechnology S.L., 18008 Granada, Spain; 3Biology Department, San Diego State University, San Diego, CA 92182-4614, USA

**Keywords:** greenhouse gases, methanotrophic communities, PGPR, *Triticum aestivum*, water stress, biostimulants

## Abstract

The search for methanotrophs as plant-growth-promoting rhizobacteria (PGPR) presents an important contribution to mitigating the impact of global warming by restoring the natural soil potential for consuming methane while benefiting plants during droughts. Our in silico simulations suggest that water, produced as a byproduct of methane oxidation, can satisfy the cell growth requirement. In addition to water, methanotrophs can produce metabolites that stimulate plant growth. Considering this, we proposed that applying methanotrophs as PGPR can alleviate the effect of droughts on crops, while stimulating atmospheric methane consumption. In this work, we isolated a series of methanotrophic communities from the rhizospheres of different crops, including Italian sweet pepper and zucchini, using an atmosphere enriched with pure methane gas, to determine their potential for alleviating drought stress in wheat plants. Subsequently, 23 strains of nonmethanotrophic bacteria present in the methanotrophic communities were isolated and characterized. We then analyzed the contribution of the methane-consuming consortia to the improvement of plant growth under drought conditions, showing that some communities contributed to increases in the wheat plants’ lengths and weights, with statistically significant differences according to ANOVA models. Furthermore, we found that the presence of methane gas can further stimulate the plant–microbe interactions, resulting in larger plants and higher drought tolerance.

## 1. Introduction

Climate change is linked to the rise of average global temperatures, which is driven by anthropogenic greenhouse gases (GHG). The consequences of climate change include dramatic changes in rainfall, floods, and droughts, along with many other costs to human life [1]. While carbon dioxide (CO_2_) receives the most attention as a global warming factor, there are other gases to consider, including methane (CH_4_), nitrous oxide (N_2_O), and black carbon [2]. The notable and increasing role of current and future emissions of methane in global warming has recently been recognized, as methane is the cause of at least one-fourth of the current gross warming [3]. Atmospheric concentrations of methane are rising rapidly, principally due to anthropogenic contributions, with wastewater treatment facilities, landfills, and livestock considered to be the key producers. The removal of atmospheric methane is needed to offset the steady release of methane, thereby limiting the contribution of this potent greenhouse gas to climate change [3]. Methane sinks occur due to chemical reactions in the atmosphere, as well as those in soils, via the action of methane-oxidizing microorganisms (also known as methanotrophs). This recognition has resulted in the advancement of cutting-edge developments and schemes to reduce the release of methane from most major contributors to emissions [4].

Implementing the natural potential of methanotrophs to offset methane emissions is of interest. Still, only minor advances have been made thus far, mostly due to relatively low levels of CH_4_ in the atmosphere. The reduction of atmospheric methane via engineered systems has been demonstrated, but the solutions are often energy-intensive or require higher methane inputs [5]. Enhancing the methanotrophic microbiomes in agricultural soils is one of the most promising solutions due to the scale of operation. Numerous studies suggest that manipulating farming practices to preserve methane sinks is possible; however, a better understanding of the interactions between methanotrophic bacteria and crops in arid environments, as well as the potential of methanotrophic bacteria as biofertilizers, is urgently needed.

In this respect, reports have noted that some biofertilizers or plant-growth-promoting rhizobacteria (PGPR) have the ability to oxidize single-carbon compounds, especially methanol. More specifically, a strain of the newly described *Methylobacterium symbioticum* species, SB0023/3^T^, has been described as a PGPR associated with spores of the arbuscular mycorrhizal fungi (AMF) *Glomus iranicum* var. *tenuihypharum* [6]. Other species of the *Methylobacterium* genus, such as *Methylobacterium oryzae*, *Methylobacterium komagatae*, or *Methylobacterium fujisawaense*, have been described as PGPR for canola plants (*Brassica campestris*) and crambe (*Crambe abyssinica*), respectively, due to their ability to produce 1-aminocyclopropane-1-carboxylate (ACC) deaminase [7,8,9]. A recent review has been published on the use of PGPR to protect plants from drought by Shaffique and coworkers [10], and another article discussed the way that microorganisms deal with water stress [11], providing more details in this respect. In general, the production of microbial biofertilizers is limited by the cost of the required nutrient media to produce a large number of microbial cells [12,13]. Using methane as the carbon and energy source for the production of biofertilizers would result in the valorization of a residue, as well as in the reduction of the required nutrient media. In addition, once the biofertilizer is used in the field, atmospheric methane would be transformed into CO_2_. We hypothesize that the stimulation of plant growth will coincide with an additional capture of CO_2_ produced during methane oxidation by the plants’ photosynthetic machinery, thereby reducing the production of the two most potent GHGs via the two mechanisms. By reducing theses GHGs, we could theoretically reduce the impacts of droughts, as these climate alterations can result in the reduction of wheat (*Triticum aestivum* L.) production by up to 20.6%, and this cereal is one of the most important sources of energy and nutrition for nearly 4.5 billion people [14].

In this paper, we explore the potential of methanotrophic bacteria as biofertilizers to improve the soil–water balance and plant growth during droughts. Here, we report on the isolation of methane-consuming microbial communities with the ability to promote wheat growth and reduce water stress in this crop under a drought stress context.

## 2. Materials and Methods

### 2.1. Metabolic Water Output Simulations

Metabolic water output simulations were carried out using previously constructed metabolic models of methanotrophic bacteria [15]. To estimate the water content of the cell biomass, a set of cultivation experiments with three methanotrophic cultures—*Methylotuvimicrobium alcaliphilum* strain 20Z^R^, *Methylococcus capsulatus* Bath, and *Methylocystis* sp. SVC1—were carried out. Strains were grown in nitrate mineral media [15]. *M. alcaliphilum* 20Z^R^ cells were cultivated with 3% and 6% NaCl to test cell-bound water at different salinities, serving as a proxy of water availability. All cultivation experiments were performed in triplicate; methane (as 20% methane headspace) was used as the carbon source. Cell cultures (1 L each) were harvested at OD = 1 by centrifugation at 4130× *g*. After centrifugation, media residues were removed, and cell biomass was weighed to obtain the wet cell weight (WCW). Cells were then frozen in liquid nitrogen and lyophilized using a Labconco freeze drying system. Cell samples were weighed again to obtain the dry cell weight (DCW).

### 2.2. Soil Samples

Soil samples of rhizospheric and nonrhizospheric soil (beige to brown clay loam; moderate, medium granular structure) were taken from various agricultural fields of Italian sweet peppers (*Capsicum annuum*) and zucchini (*Cucurbita pepo*) when fruits were ripe. In addition, a sample of soil free of plants was used as well (non-plant soil). Samples were collected from a rainfed area subject to seasonal drought at Las Gabias, Granada, Spain (37°10′55″ N, 3°41′20″ W). The soil samples were collected in plastic bags, homogenized, and sieved (using a 2 mm mesh). Then, 1 g of soil sample was immediately added to 50 mL of sterile saline solution, and resulting suspensions were serially diluted as described in Section 3.2.

### 2.3. Enrichment of Methanotrophs from Soil Samples

To identify the methanotrophic communities that proliferate under water scarcity, different soil samples were taken from a semiarid soil. Then, 1 g of each type of soil was taken and placed individually in a separate 250 mL flask containing 50 mL of sterile MSM consisting of 1 g KNO_3_; 0.2 MgSO_4_·7H_2_O; 0.02 g CaCl_2_·2H_2_O; 0.27 g KH_2_PO_4_; 0.28 g Na_2_HPO_4_; 0.01 g Na_2_EDTA; 4 mg FeSO_4_·7H_2_O; 0.6 mg ZnSO_4_·7H_2_O; 0.06 mg MnCl_2_·4H_2_O; 0.4 mg CoCl_2_·6H_2_O; 2.4 mg CuSO_4_·5H_2_O; 0.1 mg NiCl_2_·6H_2_O; 0.1 mg Na_2_MoO_4_·2H_2_O; and 0.06 mg H_3_BO_3_ [16] at a pH of 6.4 per 1 L. Each bottle was supplied with 50 mL of methane gas (99.9%; Air Liquide) to represent a 20% headspace [17]. Flasks were incubated at 30 °C with shaking at 180 rpm (Infors HT Multitron). Thereafter, 2.5 mL was taken from each flask and transferred into another 250 mL flask with 50 mL of fresh sterile MSM. Again, 50 mL of sterile methane gas was supplied to each flask, incubated at 30 °C, and shaken at 180 rpm for an additional week. This procedure was repeated up to a total of five times; in the last two passes, only 250 μL was transferred to 50 mL of fresh media. At dilution 4 and dilution 5, these split samples were designated as 1 and 2 thereafter. Therefore, the samples included the rhizospheric soil from pepper plants (RP1 and RP2), from zucchini plants (RC1 and RC2), as well as the nonrhizospheric soil close to pepper plants (SP1 and SP2) and zucchini plants (SC1 and SC2). In addition, a sample of soil free from plants was taken and labeled as “no-plant soil” (S1 and S2). A total of 2 weeks after the last dilution, 1 mL of the culture was used for DNA extraction and microbial diversity determination, 1 mL of culture was serially diluted and plated in MSM for the isolation of methanotrophs, and the rest was used for plant inoculation.

Absorbance (600 nm) measurements were collected at time point 0 and after every subsequent 12–24 h period using a Shimadzu UV-160A spectrophotometer (Shimadzu, Kyoto, Japan). For the selection of heterotrophs, trypticase soy agar (TSA) plates were used for the seeding of colonies isolated from serial dilutions of the methane-enriched cultures [18].

### 2.4. Plant-Growth Condition and Bacterial Inoculation

The surfaces of the wheat seeds (*Triticum aestivum*) were sterilized for 3 min with 5% commercial bleach (*v*/*v*) and were washed 3 times with sterile double-distilled water (H_2_Odd) for 2 h. A total of 20 seeds were placed in 0.5-L, air-tight, sealed pots that previously had been filled with 18 g of vermiculite. Pots were watered using 18 mL of water at time 0; then, the pots were sealed, and no additional water was supplied, apart from that corresponding to the inoculum. For the inoculation with the different communities, seeds were sown in the air-tight, sealed pots (VitaNtech Biotechnology, Granada, Spain) and treated with 12 mL of the liquid inoculant (consisting of a bacterial suspension from the enriched cultured on M9 buffer at an absorbance of 1_600nm_) 1 day after being sown, which represented a concentration between 1 · 10^6^ and 1 · 10^8^ of colony-forming units (CFU). Just after inoculation, the air-tight pots were sealed, and pure methane gas was supplemented (20% of the bottle headspace). The time was recorded as time 0 of the assay, and no additional water was supplied during the experiment. Seeds were incubated for 12 days under the following controlled conditions: a temperature of 20 °C and diurnal cycles of illumination consisting of 8 h with a power of 66 Watts/cm·s^2^.

### 2.5. Seed Inoculation and Plant Sampling

This experiment was designed to test the growth-promoting ability of the different methane-enriched communities on wheat seedlings. Wheat seeds were surface-sterilized as previously described [19] and were sown in air-tight, sealed pots. The following treatments were prepared: (1) for the control (non-inoculated), 3 mL 0.5× M9 buffer was added; (2) the RP community, derived from the rhizospheric soil from pepper plants; (3) the SP community, derived from the nonrhizospheric soil of pepper plants; (4) the SC community, derived from nonrhizospheric soil of zucchini plants; and (5) bulk soil, taken in the absence of any plants. The RC community, derived from the rhizospheric soil from zucchini plants, did not reach a significant level of absorbance; therefore, it was eliminated from further analysis. Each bacterial community was suspended to obtain 12 mL at an absorbance of 1 (660 nm) in 0.5× M9 buffer, which was then added to each pot. No additional water was added to the wheat seedlings until the end of the experiment. Pots were opened on day 6 of the assay to gather three seedling samples, with the aim of measuring the following parameters: shoot length, root length, fresh weight, and dry weight. The same experiment was conducted on day 12, with the aim of measuring the same parameters for the remaining samples [20].

### 2.6. Extraction of Nucleic Acids, and Next-Generation Sequencing

Nucleic acid extraction (gDNA) was performed using a FastDNA SPIN Kit for Soil (MP Biomedicals, Solon, OH, USA) and a FastPrep centrifuge. The deoxyribonucleic acid (DNA) of each biological sample was extracted in duplicate and then merged into a DNA pool, which was kept at −20 °C.

Library preparation and Illumina sequencing were carried out at the IPBLN Genomics Facility (CSIC, Granada, Spain). Amplicon libraries targeting the 16S rRNA gene were generated by a two-step PCR strategy. Gene-specific amplification was performed in triplicate, with 15 ng of gDNA in a final volume of 10 μL. Gene-specific primers, Pro341F (5′CCTACGGGNBGCASCAG3′) and Pro805R (5′GACTACNVGGGTATCTAATCC3′), were designed with Nextera overhang adapters [21]. Primers were used at a final concentration of 0.2 μM. The reaction was performed using 1× KAPA HiFi HotStart ReadyMix DNA polymerase (Roche Diagnostics, West Sussex, UK). Cycling conditions were 95 °C for 3 min, 25 PCR cycles, 95 °C for 30 s, 55 °C for 30 s, 72 °C for 30 s, and then 72 °C for 5 min. Triplicates were pooled together and validated through visualization on 1.8% (*w*/*v*) agarose gel. Amplicons were then purified using NucleoMag^®^ NGS Clean-up and Size Select Kit (Macherey-Nagel, Düren, Germany). A second PCR step attached dual combinatorial indices and Illumina sequencing adapters using the Nextera XT v2 index kit. Cycling conditions were 95 °C for 3 min, 8 PCR cycles, 95 °C for 30 s, 55 °C for 30 s, 72 °C for 30 s, and then 72 °C for 5 min. Amplicon generation was validated again through visualization on 1.8% (*w*/*v*) agarose gel and cleaned with NucleoMag^®^ NGS Clean-up and Size Select Kit (Macherey-Nagel). Concentrations were measured on the Qubit^®^ fluorometer (Thermo). Amplicons were pooled in an equimolecular manner, and the final library mix was run on a Bioanalyzer HS DNA chip to verify quality and size distribution. The library pool was then diluted and denatured, as recommended by the Illumina MiSeq library preparation guide. The 300 × 2 nt paired-end sequencing was conducted on a MiSeq sequencer.

### 2.7. Next-Generation Sequencing Postprocess

The raw sequencing data were analyzed using mothur v1.39 software [22]. First, the paired-end reads were merged into contigs and underwent quality trimming based on the avoidance of the generation of any ambiguous bases in the overlap region arising from differences in overlapping [23]. Contigs with more than 8 bp homopolymers and any ambiguous bases were removed. The remaining contigs were aligned against the full SiLVA SEED v128 database and were calculated by the k-nearest neighbor method with a k-mer size of 8, using the Needleman criterion. Sequences that failed in the alignment of the forward and reverse primer positions were removed. Then, chimerical sequences were identified using the VSEARCH algorithm implemented in mothur [24]. The non-chimerical sequences were taxonomically classified against the MiDAS database [25] and were used to construct operational taxonomic units (OTUs) through the abundance-based greedy clustering algorithm [26] using a cut-off of 97% for Prokarya. Finally, singleton OTUs were deemed to be failures and were removed.

### 2.8. Plant-Growth-Promoting Traits

#### 2.8.1. Phosphate Solubilization Assay

To find out if the strain could solubilize phosphate, SMRS 1 medium was used [27]. It contained a pH indicator that changes from purple to yellow due to medium acidification. The halo’s diameter and the brightness’s intensity indicate the phosphate-solubilizing potential of the strain used. For this test, a colony of each strain was suspended in 1 mL of 1× M9 buffer; then, 20 μL was seeded into SMRS 1 plates. The diameters of the solubilization halos were measured after 24 h of incubation in an oven at 30 °C. SMRS 1 medium was composed of: 0.5 g (NH_4_)_2_SO_4_; 0.2g KCl; 0.2648g MgSO_4_; 0.004 g MnSO_4_·H_2_O; 0.0004 FeSO_4_·7H_2_O; 0.2g NaCl; 10 g glucose; 0.5 yeast extract; 0.1 g bromocresol purple; 5 g Ca_3_(PO_4_)_2_; 18 g Bacto Agar; and distilled H_2_O up to 1 L. To prepare this medium, all of the components except the Bacto Agar and Ca_3_(PO_4_)_2_ were added, and the pH was adjusted to 7.2 with 1 N NaOH. Both the agar and the mixture were autoclaved at 120 °C for 30 min at 1 atmosphere of pressure. Finally, after cooling, the Ca_3_(PO_4_)_2_, which had been previously incubated for 16 h in the oven (Memmert, Germany) at 50 °C, was added.

#### 2.8.2. Plant Hormone Production

The production of indole-3-acetic acid (IAA) was tested using an Acquity Class I ultrahigh-pressure liquid chromatograph (UPLC) (Waters, The Netherlands) coupled to a mass spectrometer with a Xevo TQ-XS triple quadrupole analyzer (Waters, The Netherlands). For this assay, the strains were inoculated in SG medium supplemented with 50 mg/L L-tryptophan, as previously described [28].

#### 2.8.3. PCR for *mxa*F, *pmo*A, *nif*H, *nir*K, and *gst* Genes

For the PCR, the first total DNA sample was extracted from both communities and individual colonies using the DNeasy^®^ Blood and Tissue Kit (Qiagen, Venlo, The Netherlands), following the manufacturer’s instructions. The quantification of the extracted DNA was performed using a NanoDrop 2000 (Thermo Scientific, Wilmington, CA, USA). For the PCR, 30–80 ng template DNA, 1× buffer (Sigma Aldrich, St. Louis, MO, USA), 60 pmol of each primer, 0.625 U Horse-Power DNA Taq Polymerase (Canvax, Córdoba, Spain), 200 μM deoxynucleotide triphosphates (dNTPs) (Kapa biosystems, France), 2 mM MgCl_2_ (Sigma Aldrich, St. Louis, MO, USA), 0.625 μg bovine serum albumin (BSA) (New England BioLabs, Beverly, MA, USA), and 0.125 μL DMSO (Sigma Aldrich, St. Louis, MO, USA) were used. The total volume of the reaction was completed with milli-Q water up to 25 μL. The program used consisted of an initial polymerase activation phase of 4 min at 95 °C, followed by 25 cycles composed of a denaturation phase of 15 s at 95 °C, another one for hybridization for 30 s at a temperature according to each pair of primers (see Table 1), and one for elongation for 45 s at 72 °C. This was followed by a final extension at 72 °C for 7 min using an Eppendorf Mastercycler Pro S vapo thermocycler (Eppendorf, Hamburg, Germany). Primer sequences are shown in Table 1.

### 2.9. Statistical Analysis

To elucidate whether there were significant differences between treatments, statistical analysis was performed using RStudio i386, v4.0.3 software (PBC, Boston, MA (USA), 2011). First, a 95% confidence interval was applied, and the ANOVA model for the analysis of variance was obtained. Post hoc analysis was then carried out using Tukey’s HSD procedure to compare the means, two by two, thus making all possible comparisons exhaustively. The null hypothesis of equality of means was rejected when the *p*-value obtained was less than 0.05, assuming the difference to be statistically significant. Subsequently, the Bonferroni outlier test was used to check whether any of the data deviated too much from the normal distribution. If so, they were removed from the study.

## 3. Results

### 3.1. CH4-Derived Metabolic Water Output

Since methane oxidation leads to water production as a byproduct (i.e., CH_4_ + O_2_ = [CH_2_O] + H_2_O), it was speculated that methane-consuming microbes produce water intracellularly and are capable of surviving with a limited external water supply. The possibility of metabolic water to fulfill cell requirements for growth was evaluated using the flux balance model, with water integrated as one of the objective functions. Different levels of water content in the cell biomass were considered (g/g, H_2_O:DCW): 4:1, 3:1, 2:1, and 1:1. The summary of in silico and experimental data is presented in Table 2. The FBA simulation indicates that cells with a water content < 65% will release water into the environment. To further investigate the possibility of water release, we examined the water content of three strains of methanotrophic bacteria representing Type I (*M. capsulatus* Bath and *M. alcaliphilum* 20Z^R^) and Type II (*Methylocystis* sp. SVC1) methanotrophs. The water contents of the cells depended on the strain and were 74.7 +/− 2.2 for Bath, 77.4 +/− 1.9 for SVC1, and 69.5 +/− 10.17 for 20Z^R^ cells. However, in the 20Z^R^ cells grown at high salinity (6% NaCl), the water content dropped to 63 +/−1.3%. The initial data suggest that methanotrophic bacteria have the potential to fulfill most of their water requirements using methane. Thus, methane can be considered a critical but underexplored nutrient resource, especially in (semi-)arid ecosystems.

### 3.2. Enrichment Studies

Soil samples were added to MSM media, and serial dilutions were performed, as described in Section 2.3. Despite the theoretical 10^−9^ dilution generated, nearly 25 different colonies were isolated in the MSM agar plates (MSMA) and then streaked onto TSA to determine whether nonmethanotrophs were associated with methanotrophic colonies. To double check for the inability of the latest isolates to grow on methane as their sole carbon source, they were inoculated into the MSM media. Therefore, the different isolates identified in TSA seemed to be associated with growth in methane-enriched media, although they themselves were unable to grow on MSM using methane as a sole carbon source.

### 3.3. Taxonomical Characterization of Methane-Enriched Bacteria

In order to characterize the microbial diversity of the different cultures after incubation in a methane-enriched atmosphere, DNA samples obtained from each culture were used to amplify the 16S rRNA, and Illumina technology was used for metagenomic analysis. The DNA concentrations ranged from 15.0 ng/μL for the RP1 sample to 173 ng/μL for the SC2 sample. An average DNA concentration of 3.11 ng/μL was used for amplification.

Prokaryotic community structures retrieved from the different soil samples were determined by Illumina sequencing of the partial 16S rRNA gene. There were 178,409.71 raw sequences. After filtering, chimera analysis, and misaligned sequence filtering, there were 69,785 sequences. The average length of retained sequences was 374 ± 5 base pair (mean ± SD). The clustering of the sequences into OTUs resulted in 2775 different OTUs. Finally, the elimination of singletons led to 1269 OTUs.

As a result, based on this analysis, we identified that the cultures termed as S1, S2, SC1, SC2, SP1, SP2, RP1, and RP2 consisted predominantly of species associated with the *Methylocystaceae* and *Methylophilaceae* families in all cases, with representation between 38.9% for the RP2 and 65.58%, with the exception of the sample SC2, where 29.14% of the OTUs accounted for strains associated with the oxidation of methane (see Figure 1 and Table 3). In addition to the methanotrophic and methylotrophic traits identified, the enrichment cultures also produced sequences (>2%) that can be affiliated with *Chitinophagaceae*, *Comamonadaceae*, *Cytophagaceae*, and *Rhizobiaceae* families, or with the genera *Mesorhizobium*, *Pseudoxanthomonas*, *Flavobacterium*, *Brevudimonas*, *Terrimonas*, *Bosea*, *Acidovorax*, *Bradyrhizobium*, *Ferrovibrio*, *Opitutus*, and *Ferruginibacter*.

### 3.4. Plant-Growth-Promotion of the Methane-Enriched Communities

To determine whether the methane-enriched communities were useful for the promotion of plant growth, 12 mL from each community (at an absorbance of 1) was added to 18 g vermiculite containing 20 seeds of wheat (*Triticum aestivum*) and 34 mL of deionized water dispensed into 1 L, air-tight, sealed pots in triplicate. Methane gas was supplemented (20% of the headspace) every time that the bottles were opened for plant measures, and the growth of these plants was measured at 6 and 12 days after germination. The fresh and dry weights were measured. In addition, the lengths of the shoots and roots were registered. The addition of the RP1 and RP2 communities showed the highest values of root and shoot lengths, with statistical differences (*p* < 0.005) with the sample supplemented only with fresh MSM solution (in the absence of microorganisms). In contrast, samples from the S1 community showed reduced shoot and root lengths (*p* < 0.005) (see Figure 1). As the RP1 community included the largest plants (the best results in terms of plant length and plant biomass), a test was included to compare the effect of methane using a bottle containing the same conditions but not supplemented with methane. In this latest case, although plants inoculated with RP1 without methane were larger than the control in the absence of microorganisms, they were comparatively smaller than the same sample in the presence of methane.

### 3.5. PGPR Traits of Nonmethanotrophic Isolates

As nonmethanotrophs were identified, we decided to isolate them and determine whether these microorganisms were associated with the methanotrophs and whether they presented PGPR traits. In order to identify whether the nonmethanotrophs were associated with the methanotrophs, individual colonies grown on MSM agar plates were streaked onto TSA plates, and the different isolates were characterized for their ability to solubilize phosphates and for the presence of genes involved in nitrogen fixation (*nif*H coding for the nitrogenase), nitrite reductase (*nir*K), and glutathione reductase (*gst*). To eliminate the possibility of their involvement in methane oxidation, we screened for the presence of genes involved (*pmo*A coding for the beta subunit of the methane monooxygenase, and *mxa*F coding for the pyrroloquinoline quinone-dependent methanol dehydrogenase genes) using PCR on 23 isolates of the different communities (see Appendix A). None of them amplified the *pmo*A gene, but 8 out of the total 23 showed amplification with the primers specific for *mxa*F, which is involved in the second step of the methane oxidation pathway. This result points to the possibility that these 8 strains corresponded to methylotrophic bacteria, some of which have been reported as establishing cross-feeding relationships with methanotrophs, as they are unable to use methane as a carbon source, but they are able to degrade methanol and formaldehyde, which are compounds derived from the first and second steps of methane oxidation. This way they could be involved in preventing the accumulation of such metabolites, avoiding a toxic effect over the methanotrophs [33].

On the other hand, these isolates were used to determine whether they produced the plant hormone IAA. As a result of these analyses, we observed that the production of IAA was detected in 16 of the 23 isolates (Table 4).

### 3.6. Characterization of the Single-Carbon Metabolism in Methane-Enriched Cultures

DNA extracted from each methane-enriched community was used to characterize the presence of the *pmo*A and *mxa*F genes. The samples of DNA isolated from the different communities were amplified with primers for the *pmo*A and *mxa*F genes. The DNA extracted from communities S1, SC1, SC2, SP1, SP2, and RP2 resulted in *pmo*A amplification, while only the DNA extracted from the S1, SP1, and RP1 communities resulted in *mxa*F amplification. As *pmo*A genes were found in most communities, we inferred that these results confirmed the presence of methanotrophs in most types of communities (see Appendix A).

However, when PCR was performed on the DNA samples from the 23 colonies grown on TSA, no amplification was observed for the *pmo*A gene, pointing to the fact that the microorganisms isolated in TSA were unable to oxidize methane (see Appendix A). Therefore, the 23 isolates appeared to be associated with methane-oxidizing bacteria, but they themselves were not involved in methane oxidation.

### 3.7. Methane Preserves Soil Water-Holding Capacity

To determine whether the different methane-enriched communities could preserve (or even increase) the amount of moisture due to the methane-oxidation process, the humidity of the vermiculite used as a substrate was added to the water content of the plants, expressed as a percentage of the total water added (Table 5).

The highest values of relative humidity were detected in the RP2, SP2, and RP1 samples, with values of 72.29, 68.55, and 62.26%, respectively, coinciding with the largest plant samples. It is noteworthy that the RP1 sample, in the absence of methane, drastically reduced its relative humidity by almost half. These results suggest that the RP1 community could efficiently preserve water using methane-derived metabolic resources (e.g., water itself or organic molecules that trap moisture).

## 4. Discussion

In the fight against global warming and climate change, a reduction in methane and CO_2_ gases from the atmosphere is urgently needed. This reduction can only be achieved through natural processes, such as microbial communities that support active methane sinks in terrestrial soil ecosystems. Semiarid environments represent a unique resource in that respect, as they are known to withstand droughts and are established as strong sinks of methane [34,35,36]. In this work, we report the isolation of microbial communities grown in a methane-enriched media and the way these communities can interact in different manners with the development and growth of wheat. Previous studies have shown the consumption of methane by methanotrophs associated with duckweeds, resulting in high methane-oxidation activity [37]. The selection of the appropriate community, such as RP1, RP2, S1, S2, and SC2, can increase wheat plant growth and reduce the impact of droughts, as higher humidity is found in substrates containing such communities as well (see Figure 2).

The fact that none of the DNA extracted from the different isolates amplified the *pmo*A gene, but 8 out of the total 23 showed amplification with the primers specific for *mxa*F, indicates that these isolates may be methylotrophic instead of methanotrophic, using the methanol produced by other organisms. The *mxa*F gene codes for the α subunit of the methanol dehydrogenase, which is present in all methylotrophs. On the other hand, in methane-enriched cultures, the DNA extracted from communities S1, SC1, SC2, SP1, SP2, and RP2 resulted in *pmo*A amplification, while only the DNA extracted from the S1, SP1, and RP1 communities resulted in *mxa*F amplification, pointing again to the fact that methylotrophs are overrepresented over the methanotrophs in these communities. This could be the result of the extremely high methane concentrations added to the samples which, in the presence of some metals such as copper, can be spontaneously converted into methanol.

The humidity in the presence of the RP1 community was even higher in the presence of methane, revealing the importance of methane metabolism in the mitigation of droughts. This interaction of microorganisms with plants could be indirectly effected by the presence of nonmethanotrophs in the methane-enriched microbial communities.

The isolation of methane-oxidizing clones can be unfeasible, as they sometimes establish symbioses that impede their individual isolation or they can lack appropriate culturing conditions. Therefore, the isolation of communities represents an advantage over the isolation of individual strains [38,39]. It has been reported that some microorganisms establish symbiotic relationships with methanotrophs, as they can directly metabolize the CO_2_ generated as the final product of methane oxidation. Nonmethanotrophs may stimulate the growth and activity of methanotrophs via the production of additives, such as cobalamin [40].

The identification of PGPR traits, such as phosphate solubilization, IAA production, or the presence of genes involved in *gst* production in the nonmethanotrophs isolates, points to a direct or indirect role of these microorganisms in the promotion of wheat plant growth. However, the increased growth of the wheat plants with the RP1 community when methane is supplied indicates that the methanotrophs provide essential but yet-to-be established resources that support plants. Several interactions can be predicted. Methane oxidation results in CO_2_, which wheat plants then consume. The genome-scale flux balance simulations suggest that methanotrophic bacteria can fulfill their growth requirements using just methane as an energy, carbon, and water source. Thus, the methanotrophic microbiomes do not compete with plants for critical resources. A number of additional interactions can be predicted. It has been well-described that terrestrial plants release methanol through their root systems and that this metabolite can be consumed by methanotrophs [41,42,43], which in return, produce beneficial metabolites for the plant, including plant hormones for the improved development of the plant, such as auxins [44], zeatin [45], or cytokinins [46]. Some microorganisms, such as pink-pigmented facultative methylotrophs (PPFMs), have been noted for their ability to protect plants from abiotic stresses such as heat and cold [47,48] by inducing a systemic resistance to counterbalance the stresses and, in addition, to increase photosynthetic activity in crops [49]. Nevertheless, CO_2_ promotes the metabolisms of γ-proteobacterial methanotrophs via the Calvin cycle [50,51].

Cultivating methanotrophs using polluting emissions from different industries could potentially reduce the concentration of methane released into the atmosphere. However, the presence of other gases that accompany the methane in these emissions, such as carbon dioxide (CO_2_), nitrous oxide (N_2_O), fluorinated gases, including HFCs, PFC, SF6, hydrogen sulfide, etc., could have a counterproductive effect on the growth of these microorganisms, preventing their development [52]. Therefore, it is necessary to propose a study similar to the one depicted here, but one which uses the mixture of gases that are emitted by the most polluting industries for the microbial enrichment of the community. The presence of other accompanying gases may affect the resulting community composition and metabolic reactions. Therefore, additional studies with real gas mixtures are needed in addition to this one. Obtaining such communities would reduce the emission of pollutant gases by using them as a source of nutrients. In addition, the selection of plant-biostimulant methanotrophs would permit the development of plant cover that harbors methane-oxidizing communities in its root system to mitigate the methane and CO_2_ emissions from especially polluting industries, such as rice fields and landfills [53,54,55].

## 5. Conclusions

The enrichment of microorganisms in a medium with methane as the only carbon source can give rise to communities with the capacity to promote the growth of wheat. This stimulation may come from the presence of microorganisms with PGPR activity present in these enrichments that, without having the capacity to oxidize methane, do enhance it, promoting activity in the presence of methane. On the other hand, the presence of methane and its metabolism by methanotrophs seems to result in a higher degree of moisture in the vermiculite used as a substrate for plant growth. The growth of microbial communities in environments rich in methane can result in the stimulation of plant growth, with this being greater in the presence of methane, allowing for the valorization of this gas with a potent greenhouse effect. However, these microbial communities may sometimes play a counterproductive role in the growth of wheat, which is why a prior study of the community and its interaction with the plant is necessary before its use in the field.

Overall, the present study uncovered the highly positive impacts of methanotrophic bacteria on plants. Uncovering the mechanisms behind those interactions is critical for understanding the functioning of natural systems, obtaining feedback on GHG emissions, and discovering novel pathways for accelerating the adaptation of crops to climate change.

## Figures and Tables

**Figure 1 plants-12-02487-f001:**
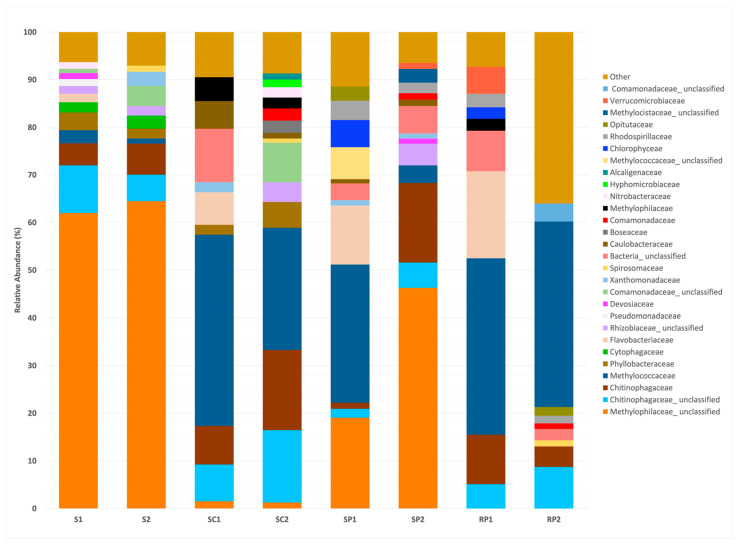
**Family-level NGS analysis results from microbial diversity in different methane-enriched communities.** Cultures termed as S1 and S2 corresponded to bulk soil, taken in the absence of any plants; SC1 and SC2 were derived from nonrhizospheric soil of zucchini plants; SP1 and SP2 corresponded to those derived from the nonrhizospheric soil of pepper plants; and RP1 and RP2 corresponded to the community, derived from the rhizospheric soil from pepper plants. Samples were designated as 1 and 2 for dilution 4 and dilution 5, respectively.

**Figure 2 plants-12-02487-f002:**
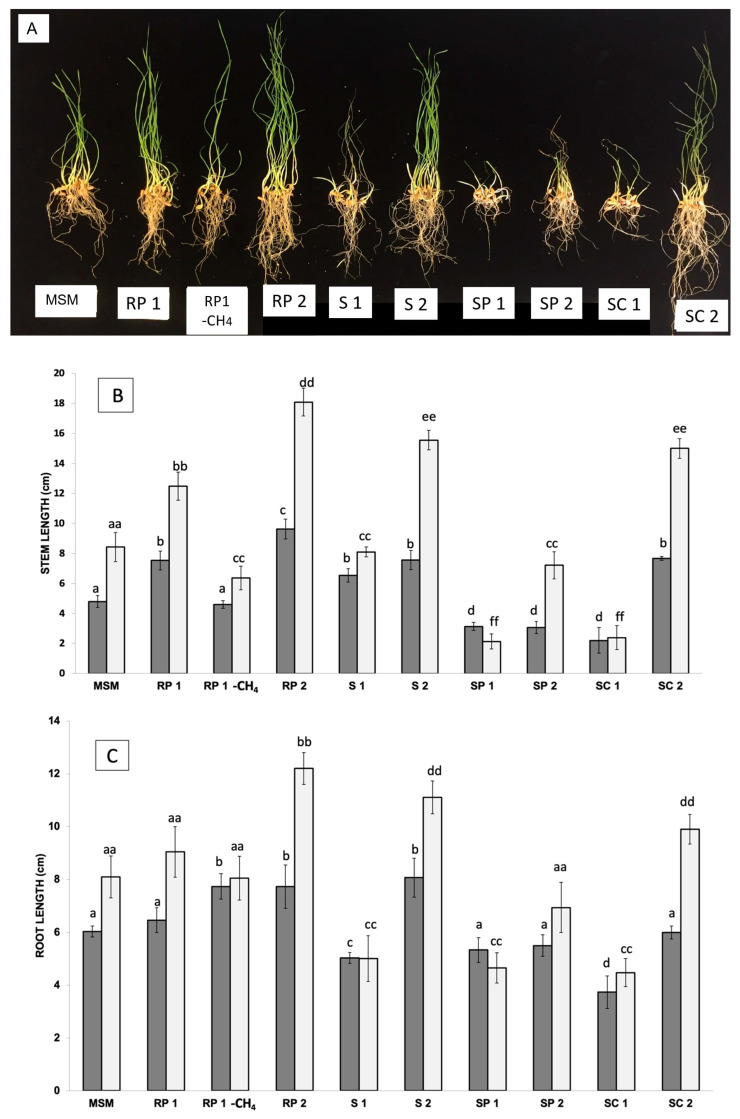
**Wheat plants treated with methane-enriched communities:** Appearance of the plants inoculated with the different communities of methane-oxidizing bacteria at 12 days after sowing (**A**). The length of the stem (**B**), the length of the root (**C**), the fresh weight (**D**), and the total dry weight of the plants (**E**) are represented for each group of plants at 6 days (represented in dark gray) and at 12 days (in light gray). Significant differences are indicated by the presence of letters, with one letter corresponding to a period of 6 days, and with two letters corresponding to a period of 12 days.

**Table 1 plants-12-02487-t001:** Primers used for the amplification of *mxa*F, *pmo*A, *nif*H, and *nir*K genes in this study.

Primers	Sequence 5′→3′	Target Gene	Hybridization Temperature	Reference
F1003	GCGGCACCAACTGGGGCTGGT	*mxa*F	60 °C	[29]
R1561	GGGAGCCCTCCATGCTGCCC			
A189gc	GGNGACTGGGACTTCTGG	*pmo*A	55 °C	[30]
mb661	TGCGAYCCSAARGCBGACTC			
polF	TGCGAYCCSAARGCBGACTC	*nif*H	55 °C	[31]
polR	ATSGCCATCATYTCRCCGGA			
Nirk-F-Brady 96	GACGAGAAGGGCAATTTC	*nirK*	58 °C	[32]
Nirk-R-Brady 96	ACTTGCCTTCGACCTTGAA			
Gst_f	CTGGAAGGCCAAGACCAAC	*gst*	56 °C	[32]
Gst_r	ACCAGATCTTGACCGAGG			

**Table 2 plants-12-02487-t002:** In silico prediction of water output in a methanotrophic organism grown with methane as the sole source of carbon and energy.

Water Content (%)	H_2_O:DCW (g/g)	Ex_H_2_O_e Flux ^#^
80%	4:1	−20.71
75%	3:1	−10.98
67%	2:1	−1.24
50%	1:1	8.49

^#^ Ex_H_2_O_e represents water exchange as μmol per h^−1^ per g cells Negative flux indicates input from the environment, and positive flux indicates secretion. The FBA analyses predict that cells will produce and secrete water if the cellular water content is below 65%.

**Table 3 plants-12-02487-t003:** Relative abundance of taxa in the different samples.

Sample	Relative Abundance (%)	Taxonomy (Genus)	Confidence (%)
**S1**	47.3 14.7 5.65 4.61 4.37 2.77 2.62 2.10 1.80 1.62 1.46 1.40 1.26 1.12 0.9	*Methylophilaceae_* unclassified *Methylophilaceae_* unclassified *Chitinophagaceae_* unclassified *Terrimonas* Uncultured Fam. *Chitinophagaceae* *Methylobacter* *Mesorhizobium* *Cytophagaceae* *Flavobacterium* *Rhizobiaceae_* unclassified *Pseudomonas* *Bradyrhyzobium* *Devosia* *Mesorhizobium* *Comamonadaceae_* unclassified	92 100 100 99 100 62 97 100 100 52 100 66 100 75 100
**S2**	47.7 16.8 6.53 5.53 4.23 2.71 2 1.90 1.29 1.14 1.08 1.07 0.96	*Methylophilaceae_* unclassified *Methylophilaceae_* unclassified *Terrimonas* Uncultured Fam. *Chitinophagaceae* *Comamonadaceae*_ unclassified *Cytophagaceae* *Rhrizobiaceae*_ unclassified *Pseudoxanthomonas* *Dyadobacter* *Mesorhizobium* *Methylobacter* *Pseudoxantomonas* *Mesorhizobium*	92 100 99 100 100 100 52 100 100 97 62 100 75
**SC1**	40.1 11.2 7.7 7.2 5.3 5.01 4.74 2.12 2.09 1.54 1.53 1.09 0.9	*Methylobacter* Bacteria_ unclassified Uncultured Fam. *Chitinophagaceae* *Terrimonas* *Flavobacterium* *Methylophilus* *Brevundimonas* *Pseudoxantomonas* *Mesorhizobium* *Flavobacterium* *Methylophilaceae_* unclassified *Caulobacter* *Terrimonas*	62 97 100 99 100 100 100 100 75 100 92 76 97
**SC2**	25.6 15.18 11.47 8.3 4.15 3.03 2.83 2.60 2.58 2.56 2.28 2.18 1.59 1.32 1.26 1.26 1.15 1.03 0.9	*Methylobacter* Uncultured, Fam. *Chitinophagaceae* *Terrimonas* *Pseudoxantomonas* *Rhrizobiaceae_* unclassified *Terrimonas* *Mesorhizobium* *Bosea* *Mesorhizobium* *Acidovorax* Methylophilus *Bradyrhizobium* *Hypomicrobium* *Terrimonas* *Methylophilaceae_* unclassified *Achromobacter* *Caulobacter* *Chitinophaga* *Dyadobacter*	62 100 99 100 52 98 75 100 97 95 100 66 100 97 92 98 76 100 100
**SP1**	28.98 17.56 12.38 6.7 5.71 4.07 3.5 2.93 1.86 1.51 1.27 1.16 0.9	*Methylobacter* *Methylophilaceae*_ unclassified *Flavobacterium* *Methylococcaceae*_ unclassified *OPB56_ge* *Ferrovibrio* Bacteria_ unclassified *Opitutus* *Chitinophagaceae*_ unclassified *Methylophilaceae*_ unclassified *Terrimonas* *Stenotrophomonas* *Brevundimonas*	62 92 100 100 100 99 97 100 100 100 99 100 100
**SP2**	25.4 20.9 14.15 5.77 5.29 3.69 3.28 2.89 2.57 2.2 1.34 1.34 1.30 1.27 1.1 1.06	*Methylophilaceae*_ unclassified *Methylophilaceae*_ unclassified Uncultured Fam. *Chitinophagaceae* Bacteria_ unclassified *Chitinophagaceae*_ unclassified *Methylobacter* *Rhizobiaceae*_ unclassified *Methylocistaceae*_ unclassified *Terrimonas* *Ferrovibrio* *Brevundimonas* *Acidovorax* *OPB35_soil_group* *Rhizobiaceae*_ unclassified *Pseudoxanthomonas* *Devosia*	92 100 100 97 100 62 52 100 99 99 100 95 100 52 100 100
**RP1**	37 18.3 10.4 8.5 5.09 4.75 2.83 2.47 2.45 0.9	*Methylobacter* *Flavobacterium* *Terrimonas* Bacteria_ unclassified Uncultured, Fam. *Chitinophagaceae* *Prosthecobacter* *Ferrovibrio* *Methylophilus* *OPB56_ge* *OPB35_soil_group_* unclassified	62 100 99 97 100 99 99 100 100 100
**RP2**	30 8.9 5 3.8 2.5 2.37 2.32 2 1.83 1.6 1.28 1.22 1.18	*Methylocistaceae*_ unclassified *Methylocistaceae*_ unclassified *Chitinophagaceae*_ unclassified *Comamonadaceae*_ unclassified *Chitinophagaceae*_ unclassified Bacteria_ unclassified *Ferruginibacter* *Chitinophagaceae* *Opitutus* *Ferrovibrio* *Emticicia* *Chitinophagaceae*_ unclassified *Acidovorax*	100 92 100 99 100 97 98 92 100 99 100 100 95

**Table 4 plants-12-02487-t004:** Phosphate solubilization, indole acetic acid production, and genes involved in PGPR traits of the different isolates. (NA stands for “not available”).

Strain	Ref. No.	Phosphate Solubilization (mm)	Indole Acetic Acid (ppb)	*mxa*F	*pmo*A	*nif*H	*nir*K	*gst*	Community of Origin
**SWW**	1	-	76.88	-	-	-	-	-	S2
**SYW**	2	18	-	-	-	-	-	+	S2
**SY**	3	17	-	-	-	-	-	+	S2
**SMA**	4	-	22.61	-	-	-	-	+	S2
**SBW**	5	-	94.18	+	-	-	-	+	S2
**SYO**	6	27	-	-	-	-	-	+	S2
**SWB**	7	29	-	-	-	-	-	-	S2
**SWO I**	8	-	-	-	-	-	-	-	S2
**SBB**	9	30	-	-	-	-	-	-	S2
**SWO II**	10	-	-	-	-	-	-	-	S2
**YCR**	11	9	500.50	+	-	-	+	-	SC2
**GCR**	12	36	147.55	-	-	-	-	-	SC2
**CCR**	13	-	660.50	NA	NA	NA	-	NA	SC2
**SRMP**	14	15	180.58	-	-	-	-	-	SRM
**SRMW**	15	11	36.31	+	-	-	-	-	SRM
**SRME**	16	12	191.79	-	-	-	-	-	SRM
**SI**	17	13	105.38	+	-	-	-	-	S1
**SCI**	18	18	108.32	+	-	-	-	-	SC1
**SCII**	19	19	113.51	-	-	-	-	-	SC2
**SPI**	20	20	127.84	+	-	-	-	-	SP1
**SPII**	21	17	105.52	-	-	-	-	-	SP2
**RPI**	22	19	137.21	+	-	-	-	-	RP1
**RPII**	23	15	212.72	+	-	-	-	-	RP2
***P. putida* KT2240**		18	4720.13			NT			

**Table 5 plants-12-02487-t005:** Humidity of the vials containing plants and methane-enriched communities.

Sample	Wet Vermiculite Weight (WVW)	Dry Vermiculite Weight (DVW)	Vermiculite Water Content (VWC = WVW − DVW)	Plant Fresh Weight (PFW)	Plant Dry Weight (PDW)	Plant Water Content (PWC = PFW − PDW)	Total Water Content (TWC = VWC + PWC)	Relative Humidity (%)
S1	33.94	17.5	16.44	0.75	0.37	0.38	16.82	36.57
S2	33.87	17.1	16.77	1.87	0.51	1.36	18.13	39.42
SC1	33.94	16.9	17.04	1.20	0.48	0.71	17.76	38.60
SC2	36.98	17.7	19.28	1.77	0.52	1.25	20.53	44.63
SP1	34.02	16.8	17.22	1.09	0.48	0.61	17.82	38.75
SP2	47.72	17.3	30.42	1.61	0.50	1.11	31.53	68.55
RP1	43.91	16.5	27.41	1.80	0.58	1.23	28.63	62.25
RP2	49.21	17.4	31.81	2.00	0.56	1.45	33.26	72.30
RP1-CH4	30.99	17	13.99	1.13	0.39	0.74	14.73	32.01
MSM	39.93	17.7	22.23	1.48	0.55	0.93	23.16	50.36

## Data Availability

MDPI Research Data Policies.

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
