# Peer review of "Isolation of Methane Enriched Bacterial Communities and Application as Wheat Biofertilizer under Drought Conditions: An Environmental Contribution"

_plants, 2023, doi:10.3390/plants12132487_

Round 1
Reviewer 1 Report
Many thanks to the authors for their efforts but I have a few questions/comments:
Title
-The title indicates the isolation of methane enriched communities. Three types of methanotrophs bacteria have been mentioned:
Methylotuvimicrobium alcaliphilum strain 20ZR, Methylococcus capsulatus and Methylocystis sp. SVC1
Did you isolate and identify these strains?
Abstract
-Lines 18, 19: “different crops including Italian sweet 19 pepper, corn and zucchini”. However, nothing related to corn was mentioned in the entire manuscript?
Introduction
-It is possible to enrich the introduction by mentioning the importance of wheat and its vulnerability to drought
Material and methods
-Lines 77, 78: The author reported on testing cell-bound water at different salinities (3, 6 %), but the results listed in Table 5 did not indicate the effect of salinity.
Table 1:
Primers for gst genes are missed
Figure 1:
Correct label for the first column “POPS” to “MSM”
Figure 1: D, E represents fresh weight and dry weight. Determine at which time of sampling (sixth or twelfth day)? And add figures and data for the other time of sampling
Results
-Lines: 297, 298; the author stated “samples from the S1 and S2 communities showed reduced shoot and root lengths (p<0.005) (see figure 1). But looking at Figure 1, we find that S2 showed enhanced shoot and root lengths compared to MSM treatment?
-The non-methanotrophic isolates; none of them amplified the pmoA gene, but 8 out of the total 23 showed amplification with the primers specific for mxaF. On the other hand, in methane-enriched cultures; The DNA extracted from communities S1, SC1, SC2, SP1, SP2 and RP2 resulted pmoA amplification, while only the DNA extracted from S1, SP1 and RP1communities resulted in mxaF amplification (lines :318-352).
According to these results, how can you conclude that the group of 23 isolates is non-methanotrophic, while the other communities are classified as methanotrophs?
-The experiments were designed in the presence of methane, but in reality, methane is mixed with other gases, especially in the emissions. This may result in reactions that are completely different from what is included in this manuscript. How do you discuss that?
Discussion
Could be improved
References
Add Details of reference no. 12 in the list
-Lines 94-98: add reference for MSM
-Add a reference to part 2.5. “Collection of biomass samples, extraction of nucleic acids and next-generation sequencing”
-Give numbers for references in table 1, add references details in the list
-Line 334: “Deng et al., 2016’, Give number for the reference, add references details in the list
Typo errors
Line 21: correct “methanotropic” to “methanotrophic”
Line 181: correct “medium” to “buffer”
- Some of the names of the organisms are not written in italics
-Some references are written in capital letters or italics for no reason
The attached file may be helpful

Author Response
Dear Reviewer, we appreciate your indications, and proceeded to introduce them in the manuscript.
With regards to the Methylotuvimicrobium alcaliphilum strain 20ZR, Methylococcus capsulatus Bath, and Methylocystis sp. SVC1, they were previously isolated in our lab, and published elsewhere.
The term corn has been removed from lines 18, 19, since results from this plant weren't finnally introduced in the manuscript.
We have introduced four lines in the Introduction section to highlight the importance of wheat and its vulnerability to drought.
We apologize for the confusion. Table 2 indicates only the flux of water required to support optimal growth of methanotrophic bacteria (i.e., biomass production at µmax) at different cellular water levels (from 50% to 80%). We compared the model predictions with cellular water levels measured for three different methanotrophic species. The water contents are listed in the main text: "The water contents of cells depended on the strain and were 74.7±2.2 for Bath, 77.4±1.9 for SVC1, and 69.5±0.12 for 20ZR cells grown at 3% NaCl. However, in cells of 20ZR grown at high salinity (6% NaCl), the water content drops to 63±1.3%."
The FBA modeling tests were integrated to investigate the minimum water requirements for methanotrophic cell, while salinity tests highlight impact of low water availability on cellular water. The data presented in Table 5 are only indirectly related to the FBA data. Compiling both, we can only speculate that huminity observed in the experimental microcosms is a cumulative effect of both- low water requirements of methanotrophic cells and, possibly, methane-driven production of metabolites that improve water holding capacity.
Primers corresponding to gst have being included now.
The label in the first column in figure 1 has been corrected from “POPS” to “MSM”
Panels D and E now contain the data for both collecting times.
In response to your comment "-Lines: 297, 298; the author stated “samples from the S1 and S2 communities showed reduced shoot and root lengths (p<0.005) (see figure 1). But looking at Figure 1, we find that S2 showed enhanced shoot and root lengths compared to MSM treatment?" the Referee is correct, and the S2 community should not be included in this sentence. Therefore we have removed the term S2 from that sentence.
An explantion to the "-The non-methanotrophic isolates; none of them amplified the pmoA gene, but 8 out of the total 23 showed amplification with the primers specific for mxaF. On the other hand, in methane-enriched cultures; The DNA extracted from communities S1, SC1, SC2, SP1, SP2 and RP2 resulted pmoA amplification, while only the DNA extracted from S1, SP1 and RP1communities resulted in mxaF amplification (lines :318-352)." have been included in the Discussion section trying to improve this section.
We have added Details of reference no. 12 in the list
With reference to Lines 94-98 we also have added reference for MSM
A reference to part 2.5. “Collection of biomass samples, extraction of nucleic acids and next-generation sequencing” have been added.
References in table 1, are given with numbers as the journal requires.
Reference included in Line 334: “Deng et al., 2016’, is in the correct format now.
Typo errors have been corrected.
In addition, the manuscript has been proof-read by the English Editing service of the editorial of this Journal, to avoid languaje issues.
Thank you for helping us to make a much better article.

Reviewer 2 Report
Manuscript ID: plants-2416317
Dear authors,
This manuscript contains up-to-date, useful and interesting information for relevant researchers, and the use of the potential of methanotrophic bacteria in reducing methane gas (a positive effect in combating global warming) and providing water needed by plants (in this study, wheat) has been studied. However, this manuscript has some writing and scientific errors that should be reviewed to be worthy of acceptance by this journal. Comments are given below:
Abstract section:
P1, L 13: Correction of the word “from” to “by”.
P1, L 13: Correction of the word “mitigate” to “mitigating”.
P1, Lines 12-18: This amount of contents is unusual for the introduction and definition of the problem in the abstract. The abstract section of your article should include sections such as introduction, materials and methods, results and research findings.
(It is not necessary to titling, but the mentioned sections must be present in your abstract).
P1, Lines 18-23: In this section, the method of conducting the experiment as well as the statistical analysis, the report of the results and the final findings of the experiment are not presented correctly, please rewrite the abstract section.
P1, L 18: Correction “In this work” to “In this work,”.
P1, L 18: Correction “isolated series of” to “isolated a series of”.
P1, L 21: Correction “methanotropic” to “methanotrophic”.
P1, L 24: Try not to use the words in the title of the article in the keywords, this will help your article to be better seen by search engines on the Internet explorations.
Introduction section:
P1, L 33: Correction of the word “into” to “in”.
P1, L 33: Correction of the word “one fourth” to “one-fourth”.
P1, L 32-34: Please provide a reference for this sentence: “The notable and increasing role of current and upcoming emissions of methane in global warming is being recently recognized since methane results into at least one-fourth of current gross warming”.
P1, L 36-37: Correction of the word “emmisions” to “emissions”.
P1, L 36-37: Delete this sentence “The growth of methane emissions is one of the major drawbacks of the objectives of the Paris Agreement.”
P2, L 56: Correction of the word “associated to spores” to “associated with spores”.
P2, L 61-66: Please provide a reference for this sentence: “The stimulation of the plant growth will coincide with an additional capture of the CO2 produced during methane oxidation by the plant photosynthetic machinery, therefore reducing in a double manner the production of the two most potent GHGs”.
P2, L 68: Please remove one as …” bacteria as as biofertilizers“.
P2, L 72: Please remove the sentence from bold situation” Metabolic water output simulations “.
Materials and methods section:
P3, L 99-102: Please provide a reference for this paragraph.
P3, L 126: Correction of the word “in absence of” to “in the absence of”.
P3, L 129: Correction of the sentence “Pots were open at day 7” to “were opened on day 7”.
P3, L 131: Correction of the word “was concluded at day 12” to “was concluded on day 12”.
P3, L 134-158: Why are no references provided for the reported tests?
P4, L 177: Correction of the sentence “To find out if the strain was able to solubilize” to “To find out if the strain could solubilize phosphate”.
P4, L 178: Correction of the sentence “purple to yellow as a result of medium acidification” to “purple to yellow due to medium acidification”.
P4, L 179-180: Correction of the sentence “The diameter of the halo and the intensity of the brightness indicate the phosphate solubilizing potential of the strain used.” to “The halo's diameter and the brightness's intensity indicate the phosphate solubilizing potential of the strain used”.
P4, L 126: Correction of the word “in absence of” to “in the absence of”.
P5, L 213: In Table 1, the referencing format should be based on the Journal’s format (numbering).
Results section:
P6, L 239: Correction of the word “sugget” to “suggest”.
P6, L 240: Correction of the word “requirents” to “requirements”.
P6, L 241: Correction of the word “nutrience” to “nutrients”.
P6, L 246: Sentence correction “environment, positive indicates” environment, and positive indicates”.
P6, L 247: Correction of the word “secrite” to “secrete”.
P7, L 248: Sentence correction “predominantly in species associated to” to “predominantly of species associated with the”.
P10, L 293-294: Sentence correction “The fresh, and dry weight were measured” to “The fresh, and dry weights were measured”.
P10, L 296: Sentence correction “with statistic differences with” to “with statistical differences with”.
P10, L 297; 392; 393: Sentence correction “in absence” to “in the absence”.
P12: Why in Figure 1 D & E, the error bar and statistical letters are not used!
P13, L 333-334: The referencing format should be modified: the methanotrophs (Deng et al. 2016).
P14, L 343: Sentence correction “the different communities amplified with” to “the different communities were amplified”.
P14, L 344: Sentence correction “resulted pmoA” to “resulted in pmoA”.
P14, L 344: Sentence correction “associated to” to “associated with”.
P14, L 345: Sentence correction “to methane oxidation process” to “to the methane oxidation process”.
P14, L 356: Correction of the word “added to the the water” to “added to the water”.
Discussion section:
P15, L 372: Correction of the word “methane enriched” to “methane-enriched”.
P15, L 380: Correction of the word “microoganisms” to “microorganisms”.
P15, L 382-383; 385: Correction of the word “stablish” to “establish”.
P15, L 387: Sentence correction “as final product” to “as the final product”.
P15, L 387: Sentence correction “as final product” to “as the final product”.
P15, L 390; 408: Sentence correction, Please delete “the“: the phosphate; the CO2.
P15, L 395-396: Correction of the word “intreractions” to “interactions”.
P15, L 410: Correction of the word “Cultivate” to “Cultivating”.
P15, L 410: Correction of the word “form” to “from”.
P15, L 411-414: Please provide a reference for this sentence: “However, the presence of other gases that accompany the methane in these emissions such as carbon dioxide (CO2), nitrous oxide (N2O), fluorinated gases such as HFCs, PFC, SF6, hydrogen sulfide, etc. could have a counterproductive effect on the growth of these microorganisms, preventing their development.”.

It seems that this manuscript needs to be reviewed by a native Englishman..
Author Response
We appreciate the contributions of this Referee to the improvement of the manuscript.
We have modified the Typos mentioned in the abstract section, and have tried to include sections such as introduction, materials and methods, results and research findings, including the employed statistic analysis.
Keywords have been modified to avoid coinciding with those presented in the title (Thank you for the tip).
Reference to “The notable and increasing role of current and upcoming emissions of methane in global warming is being recently recognized since methane results into at least one-fourth of current gross warming” have being included now.
The sentence “The growth of methane emissions is one of the major drawbacks of the objectives of the Paris Agreement.” have been removed.
No reference has been included for the sentence “The stimulation of the plant growth will coincide with an additional capture of the CO2 produced during methane oxidation by the plant photosynthetic machinery, therefore reducing in a double manner the production of the two most potent GHGs” since this part of our hypothesis. We have made clear in the text that this is our hypothesis.
P2, L 72: Metabolic water output simulations “. is now in plain text (no bold is used).
A reference has been included for the paragraph at P3, L 99-102.
In Table 1, the referencing format has been rewritten according to the Journal’s format (numbering).
All different typos have been corrected.
In addition, the manuscript has been proof-read by the English Editing service of the editorial of this Journal, to avoid languaje issues.
Many thanks for your help, to make a better version of this article.
Reviewer 3 Report
In this study the role of methanotrophs to mitigate drought plant stress was investigated. Microbial communities enriched for methanotrophs were developed from soil and rhizosphere samples grown with methane as the sole carbon source. Additionally, non-methanotrophic bacteria were isolated from ehrichment cultures and characterized for PGPR traits. Results revealed that the growth of wheat plants was improved after the inoculation with the cultures enriched for methanotrophs. The moisture of the vermiculite used as a substrate for plant growth was higher using cultures enriched for methanotrophs as inoculant. Moreover, most non-methanotrophic bacterial strains have traits involved in plant growth promotion.
General comment:
The topic of the manuscript is of interest for the scientific community because to add knowledge on the application of methanotrophic bacteria to mitigate plant drought stress. However, in general the manuscript is poorly written. A recent mini-review by Shaffique et al (doi: 10.3389/fpls.2022.87062) on the microbial mitigation of drought stress in plants should be considered for the introduction section. The description of the material and methods needs to be improved as well as the results presentation. The grammar needs to be improved throughout the manuscript.
Main specific comments:
Lines 18-19. Please improve the sentence to make more clear to the reader the strategy adopted.
Line 23. The sentence is not clear
Line 30. Please improve the grammar.
Lines 61-65. Please check this part, it is not clear the relationship between the two sentences.
Line 75. Which is the strain for Methylococcus capsulatus?
Lines 85-86. Please provide more details on soil samples . How was collected rhizosphere soil? How were the plants cultivated? At which harvesting time rhizoshere soil was collected? Were the sample immediately used for the enrichment experiments or ketpt at 4° C until use?
Line 90. Please change” 2.2. Microbial Culture Conditions to “ “Enrichment of methanotrophs from soil samples”
Line 91. Please add reference for the medium.
Line 98. Please indicate the pH of the medium.
Line 110. Please specify the liquid inoculant. Were the enriched methanotrophic cultures?
Line 112. Which are the strains?
Lines 141-142. It is not clear if the primers were developed in this study, if not please write the name and the relative reference.
Line 164. Which is the release number of SILVA database?
Line 173: Why eukaryotic?
Lines 249-263. This part must be included in the material and method section.
In the subchapter 3.3 details on the sequencing results to give information on the sequencing depth for each sample should be provided, also as a supplementary table.
Please provide box plots for data presented in table 3.
Line 296. Please include p value.
Lines 298-301. Please improve this part.
Why the letters to indicate the statistical significance were not present in Figure 1D and E?
From Figure 1 seems that the enriched methanotrophic cultures SP1, SP2, and SC1 and negatively affected plant growth. Could you explain this result?
Line 219. Please clarify this sentence.
Lines 329-334 . Please clarify this part also improving the English language.
Line 346. Please specify.
The grammar needs to be improved throughout the manuscript.
Author Response
Dear Reviewer,
Thank you very much for your kind comments on the manuscript. We find them very useful, and have proveceeded to include them.
Whe have cited two recent reviews in the Introduction section as suggested. One on how to use PGPRs for the protection of plants from drought (by Shaffique and coworkers), and another one on how microbes deal with water stress.
As suggested, sentence in lines 18-19 has been modified to make the strategy adopted clearer to the reader
Sentence in Line 23 has been reformulated to make it clear.
We have rephrased Lines 61-65 to make clear the relationship between the two sentences.
Strain for Methylococcus capsulatus is Bath (now inserted in the Materials section).
More details on soil samples, including how was collected rhizosphere soil, how were the plants cultivated, at which harvesting time rhizoshere soil was collected, and that the soil sample was immediately used for the enrichment have been included.
Heading of section 2.3 has been changed from "Microbial Culture Conditions" to “Enrichment of methanotrophs from soil samples” as suggested.
For Lines 91. and 98, the reference for the medium and its pH have been added now.
For Line 110. the liquid inoculant was M9 buffer, now specified in the manuscript.
There was a mistake in Line 173 when refering to eukaryotics. Now this sentence has been deleted.
p value is now included in Line 296. Thank you for the reminder.
The information from Lines 249-263. has been moved to the material and method section.
In the subchapter 3.3 details on the sequencing results are also provided.
Figure 1 has been updated in order to include the letters for the statistic differences.
We have tried to clarify the Line 319.
We have clarified Line 346.
The manuscript has been proof read by English native speakers with knowledge in the field.
In addition, the manuscript has been proof-read by the English Editing service of the editorial of this Journal, to avoid languaje issues.
Once again, thank you for your contributions to improve this manuscript.
Reviewer 4 Report
In this Ms. The authors report the isolation of bacterial communities enriched in metanotrophs and test these communities as Plant Growth Promoting inoculants under drough stress. The main point is that since metanotrophs generate water during methane degradation, they can be useful when water is scarce. The topic is interesting and deserves research. However, I have some major criticisms to the Ms:
1. What happened with the RC communities? No results with these communities are presented.
2. In Figure 1, panels C and D lack statistic analysis. It is not possible to see whether differences in dry weight are significant. Furthermore, what is the meaning of POPS in panel 3.
3. Lines 390-392. The presence of genes of nitrogen cycle is indicated. However, you have not found copies of nifH (nitrogen fixation) among your 23 isolates and nirK (denitrification) was found only once. I do not see implication of these strains or consortia in nitrogen cycle.
4. Figure 5. In this experiment, you used closed pots in which you added 18g of vermiculite. How can you end up with more than 20 g of dry vermiculite in most samples and up to 35 g in one of them? I think that these values invalidate the experiment. Something similar occurs with total water content. You started with 46 g of water (water plus culture), ending with variable water weights (TWC), but always well below this figure. If methanotrophs generate water and the system is closed, where is the remaining water? Have you taken into account condensed water in the pot walls? How was humidity calculated?
Author Response
Thank you very much for the useful recommendations.
1.- RC community derived from rizhospheric soil from zucchini plants did not reach a significant absorbance during the enrichment process, and therefore was discarded from further analysis with plants and microbial diversity. This is now specified in the text.
2.- With regards, to the Figure 1, panels C and D now contain their corresponding statistic analysis, and therefore now it is possible to see when differences in dry weight are significant. We are sorry for the POPS labelling in panel 3, since it should appear as MSM. Now it has been replaced by the correct labelling.
3.- For Lines 390-392. The reference to the presence of genes of nitrogen cycle has been removed.
4.- You are right in this respect. An error was commited when transcribing the data to the table, and values including the paper used as weigthing trays were included. Columns corresponding to WVW and DVW are now depicted in the absence of the paper weight. The rest of the data are unaffected. With respect to the water, we refer our data to the control sample (MSM) showing that with some samples, there is an increase in the measured water. We also have to take into account that some water molecules will be incorporated into the boimolecules of the plant tissue and microorganism cells.
In addition, the manuscript has been proof-read by the English Editing service of the editorial of this Journal, to avoid languaje issues.
Once again, we appreciate your comment to solve these issues.
Round 2
Reviewer 2 Report
Dear authors,
Thank you for the corrections based on the comments.
Author Response
Thank you for your comments.
Reviewer 3 Report
After carefully reading the revised manuscript, I found it was improved, however, several important issues that have not been addressed in the revised manuscript. See specific comments below.
subchapter 2.2 is still not clear. Please specify that different soils were used for enrichment experiments. Indeed, soil from experimental fields cultivated with zucchini and pepper in Italy and soil from Spain Were the soils from Italy arid soils? Which was the soil type from Spain?
Subchapter 2.3 must be rewritten to make it clearer. I suggest starting with the procedure of the enrichment process, the medium used, the reading of the absorbance and the isolation on TSA.
Subchapter 2.5. The title is misleading.
Why in the title of subchapter 2.6 is present “Biomass collection”? There is no reference to this aspect in the text. Please provide the correct name of the primers used. In the paper of reference 21 the primers are named with the correct names.
Were the isolates recovered from TSA identified?
Table 3 must be presented as a histogram.
The title of Table 4 must be changed because phosphate solubilization and indole acetic are phenotypic characteristics whereas the others are genes involved in PGPR traits.
Dear Editors, in my previous report I suggested the rejection of the manuscript. However, I was invited to provide comments for a second revision. I found the manuscript only partially improved, in my opinion there are several important issues that must be addressed to warrant the publication. See comments for authors.
Author Response
Dear Reviewer,
Once again, thank you for caraful reading of the manuscript and for spoting the concerned issues.
With regards to your comments on subchapter 2.2 there seems to be a confusion with the name of the plant variety. The type of used peppers are called "Italian Sweet Peppers", but no experiment was peformed in Italy or any other part than Spain. All experiments were performed in the same ochand and therefore a single description for the soil is made.
With regards to subchater 2.3 you are right in the sense that there were some confusion, and we hope it is clear now.
For Subchapter 2.5. and 2.6 the titles have been changed to avoid confusion.
When naming the oligos, we used new names, since they contain slight differences with those cited in the reference 21. Therefore we prefer to use our names.
The isolates recovered from TSA were not identified, therefore no name has been included.
With regard to Table 3, we prefer not to represent is as an histogram to avoid confusion since the importance of the less of representative species isolated in TSA will be diluted, given the important role they have and that we have found in this work.
The title of Table 4 has been changed to differentiate phosphate solubilization and indole acetic from PGPR traits.
Once again, thank you for helping us making a better article.
With kind regards,
Maximino.

Reviewer 4 Report
All my queries have been resolved satisfactorily. The correcton of the mistake in weights in Table 4 and the statistical analysis now look fine
Author Response
Thank you for your comments and support
Round 3
Reviewer 3 Report
Thanks to the Authors for providing a revised version of the manuscript. The manuscript was improved; however, I still have some concerns on some points.
Regarding oligos I do not understand the meaning of Nextera in this contest and the presence of reference 21. To my knowledge, Nextera is a technology for NGS DNA library preparation, thus, if the ref 21 was cited because the primers used in this study have been modified from those described in the paper of ref 21 it should be clearly stated.
Title of table 4 could be” Phosphate solubilization, Indole Acetic Acid Production and genes involved in PGPR traits of the different isolates.
In my opinion Table 3 should be presented as a histogram. I do not understand to what “ less representative specie” the authors refers to, because they stated in the authors response that the isolates on TSA were not identified.
Some minor changes of english language are required.
Author Response
Dear Reviewer,
Once again we appreciate your contribution to the improvement of the manuscript.
1. We have modified the name of the primers used for NGS accordingly to the ones cited in:
Development of a Prokaryotic Universal Primer for Simultaneous Analysis of Bacteria and Archaea Using Next‐
Generation Sequencing. . Takahashi S, Tomita J, Nishioka K, Hisada T, Nishijima M. e105592, s.l. : PLoS ONE , 2014,
Vol. 9(8). https://doi.org/10.1371/journal.pone.0105592.
Now the appropriate reference is included as Ref 21, and the original names of the primers is also recognised.
2. Title of table 4 as been changed accordingly to the Referees suggestions.
3. Table 3 is presented as a histogram (now in Figure 1). The Referee is right in the sense that the given information is riched when presented as histogram.
With respect to the English language, we have made use of the Service offered by this editorial group for English language editing, to solve the minor issues in this respect. Please, see the attached certificate
Once again, thank you very much for your time and patience.
With kind regards,
Maximino Manzanera

Round 4
Reviewer 3 Report
Thanks to the Authors for providing a revised version of the manuscript that resulted s improved.
In my opinion Table 3 should be eliminated because data were presented as a histogram.